# Virtual Reality Education Increases Neurologic Immersion and Empathy in Nursing Students

**DOI:** 10.3390/nursrep15090336

**Published:** 2025-09-15

**Authors:** Maria Keckler, Chia-Hsiang Hsu, Paul J. Zak

**Affiliations:** 1VITaL Research Center, San Diego State University, San Diego, CA 92037, USA; mkeckler@sdsu.edu; 2Center for Neuroeconomics Studies, Claremont Graduate University, Claremont, CA 91711, USA; chia-hsiang.hsu@cgu.edu; 3Peter F. Drucker and Masatoshi Ito Graduate School of Management, Claremont Graduate University, Claremont, CA 91711, USA

**Keywords:** empathy, neuroscience, patient journey, caring, clinical education

## Abstract

**Background/Objectives**: Virtual reality (VR) is increasingly being used in educational settings, but the evidence is mixed on whether this is better for learners. This is due in part to a reliance on self-reported “liking” of the experience rather than measuring if VR more effectively improves learner engagement and conveys information. A study was designed to determine if VR would improve nursing students’ understanding of patient interactions in the clinic (*n* = 70). **Methods**: The present study measured neurologic Immersion in nursing students during a realistic patient journey in VR and in a standard two-dimensional (2D) film. After the film, participants in both conditions had the opportunity to volunteer to help other students as a measure of the behavioral impact of the experience. **Results**: The analysis showed that VR generated 60% more neurologic value than the 2D film, and, by increasing empathic concern, positively influenced the decision to volunteer. **Conclusions**: Empathy has been shown to improve patient care while reducing healthcare provider burnout, and our findings suggest that VR that sustains neurologic Immersion should have a larger place in clinician education.

## 1. Introduction

Virtual reality (VR) is increasingly being used in education and training to enhance learning. VR uses computer modeling to enable simulated interactions in an artificial three-dimensional (3D) world and requires the use of a headset. The promise of VR-based instruction is that it would make learning more compelling and enjoyable, increase information retention, and reduce the performative aspect of teaching [1,2]. Yet, this promise has not been kept. Meta-analyses of VR compared to online and in-person learning modalities typically show null or small positive effect sizes on learning outcomes [3,4]. These effects vary across educational contexts and are influenced by content quality and instructor preparedness [1,5]. A key measure of instruction quality, information recall, is not consistently improved with VR [6,7].

There are some applications in which VR instruction appears to improve understanding of material and its application. These include training for spatial orientation, procedural skills, and technical knowledge [8,9,10]. The most valuable educational uses of VR are focused on leveraging its interactive potential [11]. While VR promises to offer advantages in specific learning domains, its potential is not yet fully realized.

Several educational settings have been documented where VR appears to confer advantages to learners [12]. One of the first educational uses of VR was in medical education [13,14,15,16]. VR simulations provide healthcare professionals with safe and repeatable training environments for clinical skill development [17]. Largely missing from medical education, however, has been effective curricular integrations of technical and clinical skills with interpersonal skills, including empathy [18].

Recent systematic reviews have shown that VR may be an effective way to inculcate empathy, at least in the short term [19,20,21]. Digital technologies have been shown to promote empathy among medical students, though less consistently among practicing clinicians [22]. Similarly, VR combined with perspective-taking appeared to enhance affective empathy through perceived emotional engagement [23]. However, empathy seems to increase only in specific program types, suggesting effects vary by population and design [24]. Empathy is known to improve patient care while reducing provider burnout, suggesting that effective VR integration to promote clinician empathy is a critical educational gap [25]. While empathy is a broad term denoting the sharing of another’s emotional or cognitive state, herein we will focus on empathic concern, which is a shared emotional state that motivates a desire to help another [26,27].

Nursing programs have experimented with VR as part of their curricula [28]. VR appears to more effectively communicate patients’ lived experiences than do didactic methods of instruction [29]. Yet, most research has relied on self-reported data to evaluate VR effectiveness [30]. Self-reports are rife with biases, especially when participants use new technologies [31]. Moving away from self-report is essential if one is to understand the underlying causes of outcomes driven by new technologies, including VR [32,33].

One source of objective data on the efficacy of VR comes from measuring neural activity in participants. The brain appears to process VR experiences in a similar way to real-life experiences, and this generally enhances training outcomes [34,35,36]. Yet, VR studies that measure brain activity seldom relate these findings to observable behaviors [37]. Most neuroscience VR studies that include behavior are focused on clinical interventions to improve patient care. In these applications, the measurement of neural responses seldom provides actionable insights [38,39,40]. For example, VR typically increases physiologic arousal as measured by electrodermal activity or cardiac responses when compared to 2D experiences [41,42,43]. This may or may not explain improved information recall in learners, as the brain-to-behavior nexus is seldom examined.

Herein we take a different tack, evaluating the effectiveness of VR using a neurophysiologic measure that captures the value the brain assigns to experiences [44,45,46]. This neurophysiologic measure, named Immersion, was then related to an observable behavior in order to identify a mechanism through which VR could improve learner outcomes. Neurologic Immersion, described in Methods, positively correlates with learner outcomes, including information recall, empathy, and helping behaviors, making it an attractive neural candidate to relate to VR experiences to behavior [44,47]. We hypothesized that VR would induce greater neurologic Immersion in participants when compared to the same information presented in 2D, and, due to this response, the VR experience would be more likely to result in a prosocial behavior. We also anticipated that neurologic Immersion would increase prosociality through its effect on empathic concern.

## 2. Methods

A randomized controlled study compared the neurophysiologic and behavioral responses of nursing students learning about a patient experience using VR and 2D video. Both formats introduced the medical journey of a female patient with a chronic illness through narrative-based storytelling, focusing on her emotional and physical challenges, barriers to effective treatment, and self-advocacy. The study design integrated several measurement approaches to trace how nursing students responded to patient experiences through different response modalities.

*Participants*: Undergraduate nursing students at a large public university were invited to participate (*n* = 70; Age: M = 21.07, SD = 2.68, 68% female). Participants were ethnically diverse (52.9% Asian, 38.6% Caucasian, 14.3% Hispanic, 1.4% Native American, and 1.4% Pacific Islander) and were assigned randomly to either the VR group or the 2D group by a lab administrator who was not part of the study.

*Consent to participate*: All participants gave written informed consent prior to inclusion in the study, which was approved by the Institutional Review Board of Claremont Graduate University (#4308) following the 1964 Declaration of Helsinki and was run 24–31 October 2022. The data were anonymized by assigning participants a random alphanumeric code. All participants were actively engaged in clinical rotations during participation in the study. No one was excluded from participation.

*Stimuli*: Two versions of a patient story were produced: a standard 2D video and a 180° VR format. The 2D version was filmed using Sony FX6 digital cinema cameras and displayed on 32-inch HD monitors (1920 × 1080 resolution) positioned 24 inches from participants. The 180° VR version was captured using Insta360 Pro2 VR cameras (8K resolution) and viewed through Meta Quest 2 headsets with 1832 × 1920 per eye resolution, 90 Hz refresh rate, and 89° horizontal field of view. Both versions shared a similar narrative arc with a runtime of 5 min 23 s. The story scripting, pre-production, filming, and editing phases followed Boorstin’s scripting principles [48]. The narrative followed a chronological structure depicting four clinical interactions between a female patient in her fifties and healthcare providers. The narrative progresses from an initial emergency department visit; to a follow-up clinic consultation; to a third clinical consultation in Tijuana, Mexico, after the patient experiences multiple failed attempts to receive a diagnosis; and a follow-up visit, resulting in a final diagnosis and resolution. The film was shot in a university nursing simulation laboratory and included standard medical equipment. Professional actors portrayed the patient (female, in her fifties) and healthcare providers (male physician, age 65; female nurse, age 25). The script was reviewed by two nursing faculty members for clinical accuracy.

*Neurophysiology*: Neural activity was measured using a commercial neurophysiology platform (Immersion Neuroscience, Henderson, NV) that produced a 1 Hz data stream by applying algorithms to Rhythm+ photoplethysmography (PPG) sensors (Scosche Industries, Oxnard, CA, USA). Neurologic Immersion convolves physiologic signals from the cranial nerves that capture responses for attention and emotional resonance that accurately predict both individual and population outcomes [44,49,50]. Immersion measures the value associated with social-emotional experiences [44] and was developed by identifying neural signals that accurately and consistently predicted people’s behaviors, including prosocial behaviors as studied here [47,51]. The Immersion platform obtains baseline neural measurements for each participant before exposure to the content, and all reported values are changes from baseline in order to control for individual differences.

The Immersion platform has been in commercial use since 2017 by both businesses and research laboratories [44]. Since Immersion was designed to predict behaviors, it is an attractive candidate to capture neurophysiologic drivers of observable prosociality in a minimally invasive way. The neuroelectrical signals that make up Immersion convolve signals capturing attention to the experience one is having, associated with dopamine binding to the prefrontal cortex, and the emotional resonance of the experience, associated with oxytocin release from the brainstem [45]. We chose to measure neurologic Immersion for this study because of the well-established relationship between oxytocin and social interactions and because the Immersion platform enables the measurement of high-frequency data [52,53,54,55].

The analysis used both average Immersion during instruction and a derived variable called Peak Immersion (Peak) following previous research [56], $${\mathrm{Peak}}_{i}=\int_{t=0}^{T}\;\left(n_{it}>M_{i}\right)d_{t}$$
where $n_{it}$ is neurophysiologic Immersion for participant i at time t when instruction begins at *t* = 0 and ends at T. The term $M_{i}$ is the median value of Immersion for person i plus 0.5 standard deviation of that participant’s Immersion. Stated more simply, Peak cumulates the most valuable parts of the experience by summing the peaks of Immersion above the threshold $M_{i}$. This variable is useful because the brain seeks to return to baseline for long data collections causing participants’ averages to be similar. For this reason, Peak is a more accurate predictor of behavior than average Immersion [45].

Note that “Immersion” with a capital “I” is a term of art for the 1 Hz convolved neurophysiologic measure used here [44]. This measure is obtained from a commercial neuroscience platform available to the general public (see below). Yet, “immersion” with a lower-case “i” is often used to denote the subjective experience of VR. We call the readers’ attention to this in order to avoid confusion since neurologic Immersion is the key independent variable in the present analysis.

*Surveys*: Participants completed surveys to provide insight into their decisions, and these were completed on tablet computers. The surveys included the measurement of empathic concern using the Interpersonal Reactivity Index (IRI; [57]). The IRI has four subscales; herein, we will restrict our analysis to the empathic concern (EC) subscale, which has been associated with endogenous oxytocin release and prosocial behaviors [49,58].

In order to assess behavioral responses, participants made binary choices to participate in a forthcoming problem-solving hackathon [59,60] immediately after the video and at one-week follow-up. Figure 1 depicts the timeline of the study.

*Data availability*: The dataset is available upon request from the authors.

*Statistical Analysis*: Mean differences were analyzed using Student’s t-tests, and Pearson correlations assessed linear relationships. Differences in proportions were analyzed using chi-squared tests. A mediation analysis was conducted to examine whether empathic concern influenced the relationship between Immersion (Peak and average) and volunteering behavior. The direct effects of both Peak and average Immersion on empathic concern were estimated after normalizing the Immersion values using their z scores, followed by estimating the effect of empathic concern on volunteering. The analysis was conducted using all the data and an indicator variable for treatment type.

We also estimated the mediation model with and without demographic controls to demonstrate robustness. To assess the significance of the mediation pathways, we conducted a bootstrapped causal mediation analysis. Bootstrap methods are more appropriate than the traditional Sobel test because indirect effects typically have skewed distributions [61]. We used 5000 bootstrap resamples following standard practice in mediation analysis to ensure stable estimates [62]. This approach provides more robust confidence intervals for indirect effects compared to traditional methods when testing for mediation. We calculated the average causal mediation effects (ACME) to quantify the indirect effects of both Peak and average Immersion on volunteering behavior through empathic concern. Standard errors were estimated using HC0 heteroskedasticity-consistent estimators [63]. The hypothesis that empathy would be associated with prosociality enabled the use of one-tailed tests. A power analysis using G*Power Release 3.1.9.6 showed that a sample size of *n* = 70 would produce statistical tests with a power of 0.90 [64] using estimated size effects from related studies relating neurologic Immersion to behavior [45]. The analysis was performed in the statistical software R version 4.4.1 [65].

## 3. Results

*Neurophysiology*. There was no difference between the VR group and the 2D group in average Immersion (VR: M = 48.88, SD = 14.69; SD: M = 54.44, SD = 14.90; t(67.99) = 1.57, *p* = 0.121; Cohen’s d = −0.38, 95% CI [−0.85, 0.10]). Yet, participants in the VR group had 60% higher average Peak compared to the 2D participants (VR: M = 1011.49, SD = 746.40; 2D: M = 632.15, SD = 477.39; t(57.83) = −2.53, *p* = 0.014; Cohen’s d = 0.61, 95% CI [0.12, 1.08]; Figure 2). Neither average Immersion nor Peak varied by ethnicity or reported gender (ps > 0.16).

*Behavior*: Nearly two-thirds of participants volunteered after the class, and this did not vary by treatment type (VR: 62.9%, 2D: 65.7%; c^2^ = 0.001, *p* = 0.99). Volunteering was unrelated to average Immersion (r = 0.048, *p* = 0.695) or Peak (r = −0.001, *p* = 0.996). The correlations of EC and average Immersion and Peak were both insignificant (Immersion: r = 0.105, *p* = 0.387; Peak: r = −0.122, *p* = 0.314). However, correlation alone may not fully capture the relationship between Immersion and empathic concern.

*Mediation*: A mediation model was estimated, analyzing whether Peak and average Immersion influenced volunteering directly and through their effect on EC. The estimation showed that Peak and average Immersion were both related to EC (Peak: β =−0.001, *p* = 0.0002); Immersion: β = 0.024, *p* = 0.007) but did not have a direct effect on volunteering (ps > 0.05). Greater empathic concern, though, increased the likelihood of volunteering (β = 0.898, *p* = 0.043, one-tailed *t*-test; Figure 3). The full estimation results are in the Appendix A Table A1.

The mediation estimation was repeated with the inclusion of controls for ethnicity and gender to test for robustness. The regression coefficients for both Peak and average Immersion had the same signs and continued to be significant. The full estimation results are in Table A2 in the Appendix A. In order to assess the significance of the mediation pathways, we conducted a bootstrapped causal mediation analysis (5000 simulations) using the model with controls. The results confirmed the significant indirect effects for both Peak (ACME = −0.00012, 95% CI [−0.00027, 0.00], *p* = 0.017) and average Immersion (ACME = 0.00645, 95% CI [0.00018, 0.01], *p* = 0.014), while the direct effects remained insignificant (ps > 0.05). This confirms the impact of Peak and average Immersion on volunteering through their influence on empathic concern.

## 4. Discussion

Virtual reality is increasingly included in educational curricula with applications ranging from technical training to fostering interpersonal understanding [66]. Yet, whether this is just a fad or improves outcomes for learners has not been established [30,67]. The inconsistency of outcomes for VR instruction is due to the quality of VR production, few fully developed courses, a lack of experience using VR by teachers and trainers, and coursework that does not adapt well to the typical learn-by-doing approach for which VR is suited [1,5,16].

The contribution herein examines whether VR instruction for nurses would increase empathy and prosocial behaviors. Our results demonstrated that a patient story in VR produced a larger neurophysiologic response compared to a traditional 2D format. The higher peak Immersion induced by VR heightened empathic concern, which, in turn, increased a prosocial behavior. This demonstrated a mechanism through which VR nursing education can enhance patient care by fostering greater empathy and potentially improved job satisfaction [25,68,69].

An innovation in our methodology was to capture neurologic responses in order to objectively compare VR to 2D. Consciously reported “liking” poorly predicts behavioral outcomes and is especially fraught when asking people to evaluate new technologies such as VR [31,45,70,71]. The larger peak Immersion for the VR condition is unlikely to be due to novelty because Immersion is largely driven by emotional responses rather than novelty or surprise [44]. Nor is the high peak Immersion due to a gender imbalance [46] as the treatment and control groups had an identical number of females. Indeed, our results were robust to the inclusion of gender and age controls.

What both researchers and practitioners want to know is if and when VR education and training are better than in-person or online coursework. While subjective assessments can provide general insights, neurologic Immersion partially or fully subsumes them because it positively correlates with information recall [44] and enjoyment [72] and is comparable interpersonally. Since our results were obtained using a modest sample size and a commercial neuroscience platform, our approach is easily replicated and extended to different educational settings.

VR has been shown to increase prosociality using a variety of stimuli for adults and for children as young as preschoolers [73,74,75]. The present study extended these findings by demonstrating that VR-delivered patient narratives that produce peak Immersion positively influenced decisions to volunteer to help others. Prosocial behaviors are common in nearly every culture [76] and increase satisfaction with life and extend healthspan [55,77,78]. Experiences that increase neurologic Immersion motivate prosocial behaviors, providing a testbed to create high-impact VR films [44,47,51]. Experiences and narratives that spike Immersion also elevate mood and energy [46,47,51,56,79]. The current study extends these findings by demonstrating that VR effectively increases peak Immersion and increases volunteering through the mediation of empathic concern. Our findings suggest that similar VR experiences viewed by a general population could increase cooperation and thereby strengthen society [80]. As in previous studies, peak Immersion is a stronger predictor of behavior than is average Immersion due, in part, to the brain’s strong tendency to return to baseline, which is typically near the average Immersion level [45,47,81].

The content and structure of VR films influence behavioral responses, and this can be assessed by measuring neurologic Immersion [82]. Indeed, storytelling is among the most effective ways to sustain neurologic Immersion [44]. The present study found that narratives shown in VR generate stronger measurable emotional responses when compared to the same information in 2D. Previously, the structure of a narrative was thought to drive Immersion responses rather than the medium in which a story was shared [45]. The present study calls this into question. While we cannot fully embrace Marshall McLuhan’s claim that the medium is the message [83], our results demonstrate that the medium, along with a well-structured message, is an effective way to transmit information and influence behavior.

There are several aspects of this study that call for replication and extension. The present experiment used a moderate sample size and studied the effect of a single instructional film. We took this approach because of the time and cost to produce VR stimuli. But, as the cost of VR production and the ease of making such films fall, the effects of multiple sessions of VR on student outcomes will be easier to measure. Our approach should also be extended to a wider set of those in clinical practice, including both early-career and senior nurses, physicians, and medical technologists. Moreover, the use of a commercial neuroscience platform that obtains data from wearable sensors makes the measurement of Immersion straightforward for non-neuroscientists.

A limitation of the present study is the high proportion of participants who chose to volunteer in both the control and treatment conditions, the measure we used of prosocial behavior, which may make the findings less reliable statistically [84]. While the decision to volunteer was made in private, participants may have felt they were required to volunteer as part of the study, even while the instructions emphasized it was not. In addition, the volunteering choice may have been influenced by the personality traits of people who choose to go into a helping profession like nursing [85]. Future research examining the role of VR instruction on prosociality should explore alternative measures of helping behaviors and should include participants who are not nurses or nursing students.

More research is also needed to establish the topics best suited to VR instruction, how often instruction should be delivered, and which learners will benefit the most [86]. Yet, even at this early stage of the analysis of VR in education, our findings indicate that VR and related technologies such as augmented reality should be added to, and could eventually replace, traditional teaching methods [3]. This includes the education of clinicians, which has been shown to be most effective when instructional design aligns simulation fidelity and learner engagement [87].

An anomalous finding was the negative sign in the mediation model relating Peak to empathic concern. This was counter to the significantly higher Peak for the VR condition compared to the 2D control. Peak may be inversely related to self-reported empathic concern due to emotional regulation. Since the brain evolved to return to baseline to reduce metabolic costs, emotional experiences are often followed by a dampening response that would occur over the film’s nearly six min run time [88]. While the partial correlation between Peak and empathic concern was significant in the mediation model, the coefficient was very close to zero. Re-estimating the mediation model using standardized coefficients for Peak and average Immersion, the estimated coefficients retained their signs and significance (ps < 0.01), and the sum of these effects was small (Peak: −0.671; Average Immersion: 0.637). This indicates that both Peak and average Immersion are necessary to increase empathic concern, unlike in previous research in which Peak alone was sufficient [49,51].

Our findings suggest that the relationship between Immersion and empathic concern is richer than a linear relationship. Indeed, neither Peak nor average Immersion was directly correlated with empathic concern alone (ps > 0.30). However, their predictive role becomes clearer when modeled together, indicating that effective VR may require sustained Immersion as well as peak emotional responses in order to provoke prosocial behaviors. As education increasingly adopts VR technology, measuring Immersion provides a platform to improve outcomes for students, patients, and society as a whole.

## Figures and Tables

**Figure 1 nursrep-15-00336-f001:**
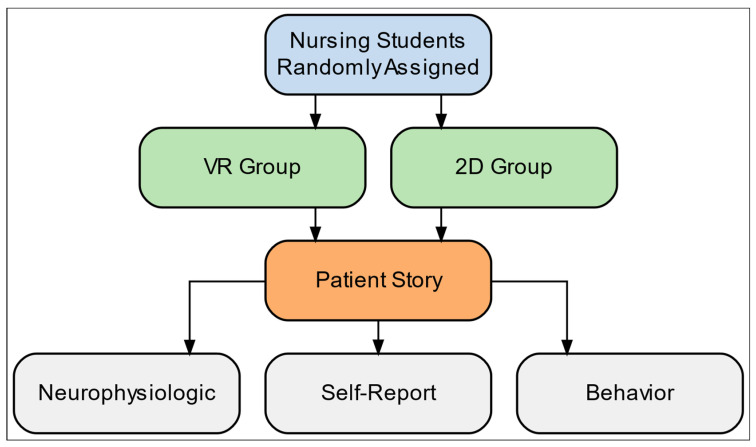
Study design and procedure showing the randomized controlled trial comparing VR and 2D patient narrative delivery with neurophysiologic, self-report, and behavioral measurements.

**Figure 2 nursrep-15-00336-f002:**
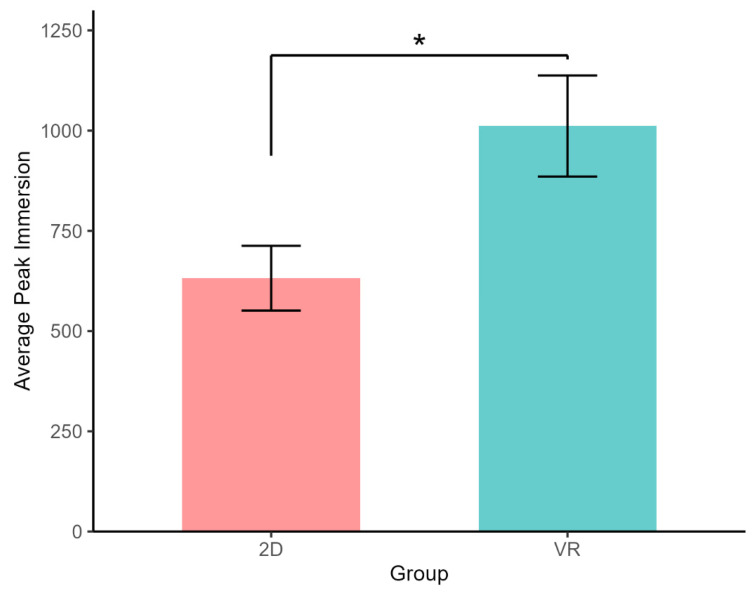
Peak Immersion was 60% higher for VR participants compared to 2D participants (*p* = 0.014, Cohen’s d = 0.61). Error bars represent standard errors. * *p* < 0.05.

**Figure 3 nursrep-15-00336-f003:**
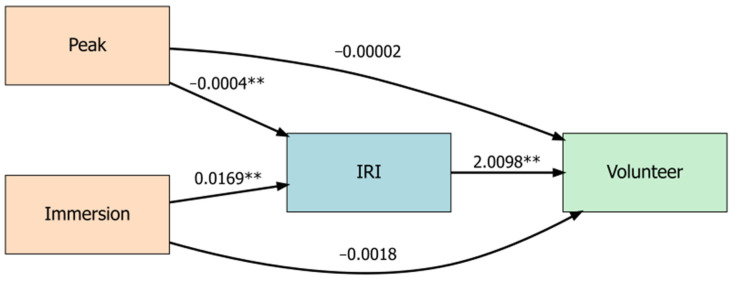
Mediation model showing the direct and indirect effects of neurologic Immersion on volunteering behavior. Both Peak and average Immersion influenced volunteering decisions through empathic concern (IRI). Coefficients are shown on paths. ** *p* < 0.01.

## Data Availability

Dataset available on request from the authors.

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
