# Peer review of "Virtual Reality Education Increases Neurologic Immersion and Empathy in Nursing Students"

_nursrep, 2025, doi:10.3390/nursrep15090336_

Round 1
Reviewer 1 Report
Comments and Suggestions for Authors
Dear Authors,
Your manuscript addresses a highly relevant and emerging area in nursing education by exploring how virtual reality (VR) can influence neurologic immersion, empathy, and prosocial behavior. The study’s innovative integration of neurophysiological measures with behavioral outcomes has the potential to contribute meaningfully to both nursing pedagogy and the broader field of health professions education. While the topic is timely and impactful, several sections of the manuscript require further clarification, methodological enhancement, and moderation of claims before it can be considered for publication. Below are detailed recommendations for improvement:
The manuscript presents a valuable and timely contribution by addressing VR’s potential to enhance neurologic immersion, empathy, and prosocial behavior in nursing education; however, several key areas require improvement. The introduction provides a solid background, yet it would benefit from integrating the most recent systematic reviews and meta-analyses (2023–2024) on VR-based empathy training to strengthen the contextual foundation. While the research design is generally appropriate, the description of the randomization process, control of potential confounding variables, and consideration of possible novelty effects should be clarified. The methods section needs more detail regarding VR/2D content standardization, validation and reliability of the Immersion Neuroscience platform, and the rationale for statistical analysis choices. The results are clearly presented, but should include effect sizes, confidence intervals, and more explanatory figure and table annotations to enhance interpretability. Conclusions are largely supported by the findings, yet some strong statements—such as VR potentially replacing traditional education—should be moderated to align with the evidence. Figures and tables are adequately prepared, though Figure 3 should display path coefficients and p-values, and all visualizations should include confidence intervals for greater transparency.
Title and Abstract
-
Consider rephrasing the title to make “Neurologic Immersion” more accessible to nursing education readers (e.g., “…Enhances Engagement, Empathy, and Prosocial Intent”).
-
In the abstract:
-
Include precise sample characteristics (n, demographics).
-
Provide effect sizes and confidence intervals alongside p-values.
-
Report non-significant findings transparently (e.g., volunteering rates).
-
Introduction
-
Expand the literature review with the most recent systematic reviews/meta-analyses (2023–2024) on VR in empathy training for nursing students.
-
Strengthen the articulation of the research gap. Specifically, the lack of studies linking neurophysiological measures to empathy/prosocial behavior in VR-based nursing education.
-
Present a clear logic chain linking VR exposure → Immersion increase → empathy enhancement → prosocial behavior.
Methods
-
Describe the randomization process in detail (method, sequence generation, allocation concealment).
-
Discuss the gender imbalance and possible influence on empathy scores.
-
Provide more detail on VR and 2D content development, including:
-
Content validation by experts.
-
Measures to ensure equivalence in narrative and quality across formats.
-
Whether participants had prior VR experience (novelty effect).
-
-
Add validation and reliability evidence for the Immersion Neuroscience platform; mention potential limitations.
-
Clarify which IRI empathy subscale was used and report Cronbach’s alpha.
-
Justify the choice of bootstrapped mediation analysis with 5000 simulations and explain why one-tailed tests were used in some analyses.
Results
-
Add effect sizes and 95% confidence intervals to tables and figures.
-
In Figure 3 (mediation model), include path coefficients and p-values directly.
-
Discuss in more depth the discrepancy between average Immersion (non-significant) and Peak Immersion (significant).
-
Address potential confounding factors affecting volunteering rates.
Discussion
-
Moderate strong statements suggesting VR could replace traditional education; instead, position VR as a complementary approach.
-
Provide a deeper analysis of the anomalous negative coefficient for Peak in relation to empathy.
-
Acknowledge possible novelty effects and the lack of long-term follow-up data.
-
Expand recommendations for future research, including cost-effectiveness analysis, multiple VR scenarios, and larger samples.
Author Response
replies attached

Reviewer 2 Report
Comments and Suggestions for Authors
Dear authors,
The manuscript's topic is very topical, as innovative technologies, including virtual reality (VR), have become inevitable in general and nursing education. Due to the specificity of the outcome, they are applied with special care. The special significance of your study is that it deals with aspects of the application of VR, such as neurological immersion, which is generally important for implementing innovative strategies in education, and empathy as a "soft skill" specific to the nursing profession.
I would like to make suggestions for improving the manuscript:
Abstract: The abstract is structured according to the journal's instructions.
Introduction. The introduction is coherent, easy to follow, provides a comprehensive overview of the issues the manuscript addresses, and follows a funnel-shaped structure.
Methods. For a more comprehensive overview of the study, it would be significant to state.
In the Participants section: How many students are enrolled in the nursing study program, in which year of study the teaching was implemented where VR was applied, How many students were in each group, did any of the students refuse to participate in the study, did the students have experience with the application of VR in teaching or was this their first experience, did they have the answer option of earlier recreational use of VR. Please clarify this.
In the part of the survey that relates to the importance of empathy, as a specifically analysed aspect of the application of VR, and the IRI as a multidimensional instrument that was applied, it is necessary to add data on the number of items, possible answers, method of scoring results, and subscales of this questionnaire.
Also, please add information about how the surveys were distributed to students.
In Figure 1, please write/add the student number for each group.
In the statistical analysis section, it is necessary to add which statistical software was used.
Results: The results are partially adequately presented. The data in the Behaviour section is difficult to track. For a clearer view, I suggest presenting them in a Table. For an adequate insight into the results indicating empathy, it would be essential to show the data for both groups in a table, by all items or at least by subscales. Please add this data.
Discussion. The discussion is extensive, and the authors draw attention to many important questions raised by their research and the results of other authors. After adding the results on the individual domains of the IRI questionnaire, it would be essential to discuss each one of them.
The findings are concise, well-argued, and based on research results. I suggest that the findings/conclusions be created as a separate section.
The references mentioned are relevant to the topic that the paper dealt with.
I hope you find my comments helpful.

Author Response
replies attached

Reviewer 3 Report
Comments and Suggestions for Authors
The article is quite interesting and well-written. It fits well within the scope of the journal. Below, I provide a few minor remarks that should be taken into consideration:
p. 1 Introduction: It would be appropriate to briefly characterize or define Virtual Reality right at the beginning, i.e., in the Introduction. One cannot assume that all readers know what Virtual Reality is.
p. 2: There is a bibliographic entry (Bracq et al. (2019)). It should be (Bracq et al., 2019).
Introduction: One of the key terms in the article is “empathy.” It would be worthwhile to provide at least a few definitions of this term, especially since there is no consensus on it in the subject literature.
p. 2, Discussion: The last sentence on page 2, which starts as follows: “All participants gave written informed…” is cut off. This is probably a technical error.
p. 6: On page 6, a sentence begins with: “Virtual reality…”. To maintain consistency, it should be Virtual Reality or VR.
At the end of the article, it would be good to mention the limitations related to the conducted research.
Author Response
replies attached

Round 2
Reviewer 1 Report
Comments and Suggestions for Authors
Please clarify randomization procedures: how the sequence was generated (software, block/stratification) and how allocation concealment was ensured (built-in randomizer/opaque sealed envelopes, who implemented it).
In Methods, state that the covariates reported in Results (gender, ethnicity) were pre-specified (or explain the rationale if they were exploratory).
Ensure 95% confidence intervals are reported alongside key estimates in the tables (not only in the text).
Add standard Data Availability and Conflict of Interest paragraphs in line with the journal’s format.
(Optional but helpful) Report internal consistency (Cronbach’s α) for the IRI subscale used in your sample.
Expand the limitations to acknowledge potential confounders for volunteering rates (prior prosocial tendencies, peer effects, order/time-of-day).
Author Response
Table and figures have been updated and improved, both figuratively and in the captions.
